# Raised FGF23 Correlates to Increased Mortality in Critical Illness, Independent of Vitamin D

**DOI:** 10.3390/biology12020309

**Published:** 2023-02-14

**Authors:** Onn Shaun Thein, Naeman Akbar Ali, Rahul Y. Mahida, Rachel C. A. Dancer, Marlies Ostermann, Karin Amrein, Gennaro Martucci, Aaron Scott, David R. Thickett, Dhruv Parekh

**Affiliations:** 1Birmingham Acute Care Research Group, Institute of Inflammation and Ageing, University of Birmingham, Level 2 Queen Elizabeth Hospital Birmingham, Birmingham B15 2TH, UK; 2Warwick Hospital, Warwick CV34 5BW, UK; 3King’s College London, Guy’s & St Thomas’ Hospital, London SE1 7EH, UK; 4Division of Endocrinology and Diabetology, Medical University of Graz, 8036 Graz, Austria; 5Department of Anesthesia and Intensive Care, Istituto Mediterraneo per i Trapianti e Terapie ad Alta Specializzazione (IRCCS-ISMETT), 90133 Palermo, Italy

**Keywords:** FGF23, critical illness, intensive care

## Abstract

**Simple Summary:**

Fibroblast Growth Factor 23 (FGF23) is a hormone which is known to control the levels of vitamin D, phosphate, and calcium in the body. There is evidence that high levels of FGF23 in patients with chronic kidney disease may increase the risk of death. The aim of this study is to see if patients who are extremely unwell (admitted to intensive care) with high levels of FGF23 are more likely to die than patients with lower levels. We investigated patients from two intensive care studies. We showed that patients who had higher FGF23 levels when they were admitted had a higher risk of death compared to patients with lower FGF23 levels. This applied to patients without chronic kidney disease as well as those with normal or low vitamin D levels. This may mean that FGF23 can affect the way our immune system works in a previously unexplored way. With more research, treating FGF23 levels might be a way to improve the survival of intensive care patients.

**Abstract:**

Background: Fibroblast Growth Factor (FGF23) is an endocrine hormone classically associated with the homeostasis of vitamin D, phosphate, and calcium. Elevated serum FGF23 is a known independent risk factor for mortality in chronic kidney disease (CKD) patients. We aimed to determine if there was a similar relationship between FGF23 levels and mortality in critically ill patients. Methods: Plasma FGF23 levels were measured by ELISA in two separate cohorts of patients receiving vitamin D supplementation: critical illness patients (VITdAL-ICU trial, *n* = 475) and elective oesophagectomy patients (VINDALOO trial, *n* = 76). Mortality data were recorded at 30 and 180 days or at two years, respectively. FGF23 levels in a healthy control cohort were also measured (*n* = 27). Results: Elevated FGF23 (quartile 4 vs. quartiles 1–3) was associated with increased short-term (30 and 180 day) mortality in critical illness patients (*p* < 0.001) and long-term (two-year) mortality in oesophagectomy patients (*p* = 0.0149). Patients who died had significantly higher FGF23 levels than those who survived: In the critical illness cohort, those who died had 1194.6 pg/mL (range 0–14,000), while those who survived had 120.4 pg/mL (range = 15–14,000) (*p* = 0.0462). In the oesophagectomy cohort, those who died had 1304 pg/mL (range = 154–77,800), while those who survived had 644 pg/mL (range = 179–54,894) (*p* < 0.001). This was found to be independent of vitamin D or CKD status (critical illness *p* = 0.3507; oesophagectomy *p* = 0.3800). FGF23 levels in healthy controls were similar to those seen in oesophagectomy patients (*p* = 0.4802). Conclusions: Elevated baseline serum FGF23 is correlated with increased mortality in both the post-oesophagectomy cohort and the cohort of patients with critical illness requiring intensive care admission. This was independent of vitamin D status, supplementation, or CKD status, which suggests the presence of vitamin D-independent mechanisms of FGF23 action during the acute and convalescent stages of critical illness, warranting further investigation.

## 1. Introduction

Fibroblast growth factor 23 (FGF23) is a bone-derived hormone that plays a key role in phosphate homeostasis, primarily in response to hyperphosphatemia. As a result, FGF23 also influences vitamin D and calcium homeostasis through interactions with vitamin D conversion enzymes and parathyroid hormone (PTH) (Figure 1). Vitamin D supplementation causes a rise in FGF23, acting as a counterregulatory hormone to reduce 25(OH)2D3 hydroxylation by reducing 1α hydroxylase (CYP27b1) activity and increase 1,25(OH)2D3 breakdown by increasing 24 hydroxylase (CYP24a1) activity. At normal physiological concentrations, FGF23 exerts its intracellular effects by forming a tertiary cell membrane complex with FGF receptors (FGFRs) and α-klotho, a co-receptor that increases FGFR binding affinity. 

Critical illness can be secondary to a variety of pathologies and underlying diseases, including those acquired after major surgery, as patients have a high risk of secondary infection. Post-oesophagectomy patients require admission to, and often a prolonged stay in, critical care units due to significant complication risks, especially pneumonia and sepsis. 

Vitamin D deficiency (VDD) is both prevalent and severe in critically ill patients and those with sepsis, and it is associated with high short- and long-term mortality [1,2,3,4]. However, clinical trials involving the supplementation of cholecalciferol in critical illness have yielded variable results [1,5,6,7,8,9]. Clinical benefit has so far only been observed in patients with severe VDD [5,10]. These clinical trial data conflict with observational data regarding the relationship between VDD and patient outcomes, suggesting the presence of uncharacterised counter-regulatory metabolic pathways that blunt the response to high-dose vitamin D monotherapy [8,11,12]. It remains unclear how critical illness affects the vitamin D metabolome and important downstream vitamin D-responsive, anti-inflammatory, and anti-infective pathways [13,14]. 

FGF23 may play a role in both the dysregulation of vitamin D metabolism and the inflammatory responses observed in critically ill patients, independent of their vitamin D status. FGF23 has been reported to exert vitamin D-independent effects, including stimulating the release of interleukin 6 (IL-6) and C-reactive protein (CRP) from hepatocytes in a murine inflammatory model [15]. In murine studies, FGF23 also drives macrophage polarisation towards a pro-inflammatory phenotype and stimulates the release of tumour necrosis factor α (TNFα) via RAS/MAPK signalling [16,17,18]. In turn, FGF23 is secreted by macrophages in response to toll-like receptor 4 (TLR4) stimulation. It has been suggested that FGF23 may also affect macrophage function, based on klotho receptor RNA expression and interleukin production [17,19]. These vitamin D-independent effects suggest a potentially broader role for FGF23 in the inflammatory and immune responses of critically ill patients, irrespective of their vitamin D status.

The clinical importance of FGF23 has been extensively studied in chronic kidney disease (CKD). In CKD patients, high serum FGF23 has been independently correlated with an increased risk of infection, resulting in hospitalisation and increased mortality [20,21,22]. Elevated FGF23 has also been independently associated with frailty and pre-frailty, with a notable correlation between raised FGF23 and mortality in older individuals [23,24]. Furthermore, increased FGF23 has been identified as a post-operative indicator of acute kidney injury and non-occlusive mesenteric ischaemia [25], suggesting that FGF23 levels may be a potential link between multimorbidity and critical care outcomes.

Aim: We aim to determine the association between circulating FGF23 levels and mortality in critical illness, independent of vitamin D status or chronic kidney disease. Demonstrating the role of FGF23 may explain the variable outcomes of vitamin D replacement trials for critically ill patients.

## 2. Methods

Data and stored samples were obtained from two randomised, placebo-controlled clinical trials and healthy control participants.

### 2.1. Critical Illness Cohort

Anonymised data were obtained from the VITdAL-ICU trial population (NCT01130181). Between May 2010 and September 2012, 475 patients were recruited from five intensive care units (ICUs) in a single large tertiary centre in Austria, as part of a randomised, double-blind, placebo-controlled trial investigating the effect of vitamin D replacement in vitamin D-deficient, critically ill patients [5]. Both medical and surgical patients were recruited. Ethics approval was granted by the Medical University of Graz, Austria (reference number 21–214 ex 09/10, EudraCT-Nr.: 2010-018798-39). Patients were randomly assigned to either a placebo or a vitamin D3 group (which received a high-loading-dose regimen of a single dose of 540,000 IU vitamin D3, followed by once-monthly 90,000 IU doses for 5 months). The database was obtained from the original investigators. FGF23 levels were analysed from blood samples collected on day 0 and day 7, which were thawed from −80 °C and analysed by ELISA. Recruitment to the translational arm was optional, so the number of patients with FGF23 measured was limited. Post critical care discharge, patient samples were not collected, leading to the loss of some day-7 (timepoint 2) FGF23 measurements. Mortality data were recorded at 180 days from recruitment.

Inclusion criteria: patients 18 years of age or older, expected to stay in the ICU >48 h, with a plasma 25-hydroxyvitamin D level of ≤20 ng/mL. Exclusion criteria: patients with severely impaired gastrointestinal function; participants of other trials; pregnant or lactating women; patients with hypercalcaemia (total calcium > 10.6 mg/dL or ionised calcium > 5.4 mg/dL), tuberculosis, sarcoidosis, or nephrolithiasis; or patients deemed unsuitable for study participation (e.g., those with psychiatric disease, those living remote from the clinic, or prisoners). 

### 2.2. Oesophagectomy Cohort

Samples were obtained from the VINDALOO trial population (ISRCTN27673620). From this, 76 patients were recruited into the multi-centre, randomised, double-blind, placebo-controlled trial in the UK. The VINDALOO trial investigated the effect of pre-oesophagectomy vitamin D supplementation on early biomarkers of acute lung injury [4]. Oesophagectomy patients experience a timed critical illness insult during major surgery, and many develop significant post-operative complications, especially sepsis and pneumonia. Serum samples (taken pre-allocation, post- drug or placebo dosing, and at day 3 or 4 post-operation (timepoint 2)) were thawed from −80 °C and analysed by ELISA. Ethical approval was granted by the South Birmingham Research Ethics Committee (REC 12/WM/0092). 

Inclusion criteria: patients undergoing a planned thoracic oesophagectomy, who are 18 years of age or older, if male, or 55 or older or more than two years post menopause, if female. Exclusion criteria: patients with known vitamin D intolerance, sarcoidosis, hyperparathyroidism, or nephrolithiasis; those taking more than 1000 IU/day of vitamin D supplementation in the month preceding enrolment; those with baseline serum corrected calcium >2.65 mmol/L; those undergoing haemodialysis; those who were pregnant or breastfeeding; those taking drugs (cardiac glycoside, carbamazepine, phenobarbital, phenytoin, primidone, long-term immunosuppressant therapy); those taking an oral preparation containing >10 micrograms of vitamin D/day up to 2 months before the first dose; or those with COPD who had an FEV1 <50% and a predicted or resting oxygen saturation of less than 92%. Mortality data were recorded at two years from recruitment.

### 2.3. Healthy Controls 

Plasma samples were collected from 27 healthy volunteers (REC ref:19/WA/0299 and REC ref: ERN 12-1184R2), who were recruited from staff and patients attending routine outpatient appointments at Queen Elizabeth Hospital in Birmingham, UK. 

Inclusion criteria: patients between 18 and 99 years old. Exclusion criteria: patients who were acutely admitted to the hospital, who concurrently used oral or inhaled corticosteroids, or who were pregnant or breastfeeding. 

### 2.4. FGF23 Quantification 

Intact FGF23 levels were measured by ELISA. Samples of critically ill patients were measured using an Intact Human FGF23 ELISA kit (Quiadel Immutopics 2nd generation kit, San Diego, CA, USA). Samples of oesophagectomy patients and healthy controls were measured using a Human FGF23 DuoSet ELISA kit (Biotechne, Abingdon, UK). Different ELISA kits were used due to regional availability and the spatial and temporal separation of the sample analysis. Results derived from different kits were deemed not comparable for this reason. FGF23 levels were determined according to the manufacturers’ ELISA protocols in each instance.

### 2.5. Statistical Analysis

FGF23 levels were not normally distributed (VITdAL-ICU Shapiro–Wilk *p* = 0.0001, VINDALOO Shapiro–Wilk *p* = 0.0001). They were log converted to produce normally distributed data for statistical analysis in SPSS v26.0.0.0 and GraphPad Prism v8.2.0. Student’s T-tests were used to determine the statistical significance between normally distributed groups. Chi-squared tests with Yates corrections and Mann–Whitney U tests were used to determine the statistical significance of non-parametric data. 

## 3. Results

The VITdAL-ICU study recruited 475 patients, 126 of whom were measured for FGF23, with laboratory error discounting the results from another three (Table 1). A total of 123 patient samples were therefore included in the analysis (66 male (54%), median age 68.0 (interquartile range (IQR) 56.0–77.0)). Longitudinal follow-up on 180-day mortality was available for all 123 patients. Of the patients with CKD, 18 were treated with placebo and 15 were treated with vitamin D. 

The VINDALOO study recruited 76 patients (64 male (84%), median age 67.0 (IQR 58.0–71.5)). Of these, only 60 patients produced two-year mortality data (Table 1). 

To compare, 27 healthy controls were recruited (11 male (41%), median age 70 (IQR 61.0–78.0)). No long-term follow-up data were collected from these patients. Patients diagnosed with CKD had stage 1, and therefore were not on dialysis.

Median FGF23 levels in healthy controls were not significantly different from those in the oesophagectomy cohort (*p* = 0.4802, Figure 2). High FGF23 values measured in the healthy control group did not correspond to the values measured in patients with CKD. 

### 3.1. CKD Affects FGF23 Levels (VITdAL-ICU Critical Illness Cohort) 

Patients with CKD had increased levels of FGF23 at baseline compared to their non-CKD counterparts (*p* < 0.05) (Table 2), and these higher levels persisted to day 7. There were no patients with CKD recruited to the VINDALOO trial. 

### 3.2. FGF23 Values Do Not Significantly Change over Illness Duration 

There was no significant difference in FGF23 levels between baseline and follow-up (timepoint 2) in either the critical illness or the oesophagectomy cohort (Table 1). 

### 3.3. Vitamin D Supplementation Does Not Affect FGF23 Levels

There were no significant differences in FGF23 level between the placebo and vitamin D-supplementation groups at timepoint 2 in either the oesophagectomy or the critical illness cohort (Table 3). Although an increase in FGF23 was noted in CKD patients, this was not significant, and adjusting for patients with CKD did not make any significant difference to acute FGF23 levels measured at the timepoint 2 follow-up.

### 3.4. Raised Serum FGF23 Is Correlated with Increased Mid- and Long-Term Mortality 

Patients with CKD were excluded from this analysis. Patients with higher FGF23 levels (top quartile of cohort) demonstrated a significantly higher mortality rate than those with lower FGF23 levels (*p* < 0.05) (Figure 3A,B). There was significant association between FGF23 levels and mortality at 180 days (VITdAL-ICU) and at two years (VINDALOO). The odds ratios for mortality in critical illness patients and oesophagectomy patients were 5.329 (*p* < 0.002, 95% CI 2.381, 11.923) and 3.000 (*p* = 0.04, 95% CI 1.014, 8.880), respectively.

### 3.5. Patients Who Died Exhibited Raised FGF23 Levels

Those patients who did not survive demonstrated significantly higher FGF23 levels at day 0 in both the oesophagectomy (1.6 times greater in patients who did not survive) , *p* < 0.05 and critical illness (9.9 times greater in patients who did not survive, *p* = 0.05) cohorts (Figure 3C). 

### 3.6. Vitamin D Levels Are Not Correlated to 180-Day or Two-Year Mortality 

No significant association was found between baseline 25(OH)2D3 levels at D0 and mortality in either cohort of patients (Figure 3D). There was no significant difference in 1,25(OH)2D3 levels at D0 vs. the two-year mark in oesophagectomy patients (*p* = 0.0740; alive median = 109.1, IQR = 35.55; deceased median = 86.25, IQR = 40.6) or in 1,25(OH)2D3 levels at D0 vs. 180 days in critical illness patients (*p* = 0.4008; alive median = 26, IQR = 48; deceased median = 28, IQR = 41.75). 

## 4. Discussion

FGF23 is implicated in the inflammatory immune response, phosphate and calcium homeostasis, and dynamic turnover in bone [18,26,27,28,29]. Experimental studies suggest a direct pro-inflammatory effect of FGF23 on macrophages, which is key to the innate immune response to infection/inflammation [27]. Evidence from patients with CKD suggests an association between raised FGF23 levels and increased susceptibility to line infections, sepsis, and mortality [30,31,32]. The FGF23 levels measured in our patient cohorts were above the normal ranges (18–180 pg/mL) previously described [33,34,35]. CKD is associated with disrupted calcium and phosphate homeostasis, and, in line with other reports, we observed that critically ill patients with CKD exhibited elevated levels of FGF23 [36]. That said, FGF23 has not been extensively investigated in the context of critical illness. This study demonstrates elevated circulating FGF23 levels in critically ill and perioperative patients as well as their association with increased mortality in two acutely ill cohorts, with similar findings. Our results a demonstrate significant association between FGF23 levels and mortality at 180 days (VITdAL-ICU) and two years (VINDALOO). This suggests that elevated FGF23 levels at baseline correspond to short-, mid-, and long-term mortality for patients who develop critical illness. 

Compared to the literature documenting FGF23 levels in healthy populations, our healthy controls demonstrated higher FGF23 levels, with no significant difference from either the critical illness or the oesophagectomy cohorts. As FGF23 is involved in vitamin D homeostasis, factors affecting 1,25(OH)2D3 can affect circulating FGF23; these include, but are not limited to, ethnicity, supplementation, and season. This is supported by the significant variability seen in this population. However, elevated FGF23 in both healthy controls and patients suggests that circulating FGF23 is not an acute phase protein. High FGF23 levels, whether pre or post inflammatory insult, may affect the regulation of the innate immune response, consequently causing poorer clearance of secondary infections and death. This is supported by the lack of a significant difference in FGF23 levels between patients treated acutely with vitamin D and those treated with placebo, suggesting a subacute, chronic effect or a predisposition to severe inflammation. 

No short-term dynamic changes in FGF23 were observed, suggesting that FGF23 either becomes elevated at the inception of illness or is pre-existing. The lack of acute changes in serum FGF23 could also be attributed to FGF23 acting in a damage-associated molecular pattern (DAMP). In the acute stages of inflammation, DAMPs are slow to rise but remain sustained, contributing to defence mechanisms and perhaps also promoting the inflammatory response. 

Investigation of molecular acute dynamic responses in critical illness is difficult due to the general unpredictability of onset. Oesophagectomy patients are therefore a more suitable cohort for examining these responses. To define the role of FGF23, better understand the mechanisms driving its production, and understand its temporal changes in critical illness, further translational study over a longer period is warranted. 

Our data, if supported by further detailed cohort studies (such as will be available upon conclusion of our trial VITDALIZE (ISRCTN 44822292)), may inform the potential of therapeutic FGF23 blockade to reduce morbidity and mortality from critical illness [37]. Burosumab, a monoclonal antibody to FGF23, is already licenced for the treatment of X-linked hypophosphataemic rickets [38] and, following appropriate investigation, could potentially be repurposed to determine whether FGF23 blockade might benefit critically ill patients.

In both our critically ill patient cohorts, patients who died had substantially elevated serum FGF23 independent of vitamin D status—and this persisted irrespective of vitamin D supplementation. Under normal physiological conditions, FGF23 tightly regulates alterations in the homeostatic balance of phosphorus, 25(OH)2D3, and 1,25(OH)2D3, in turn, causing a negative feedback loop. The previous literature has observed a correlation between VDD and worse outcomes in critical illness, and has even shown critical illness itself to cause relative vitamin D deficiency [10,39,40]. The observed association between FGF23 and mortality was independent of vitamin D status, suggesting that FGF23 may be a mechanistic driver; perhaps deranged vitamin D is a consequence rather than a cause. 

It is possible that FGF23 drives inflammatory and immunoregulatory pathways in critical illness, independent of vitamin D status. This is supported by the observation that both cohorts demonstrated an association between FGF23 and mortality, which perhaps indicates a unifying pathway of inflammation driving progression to critical illness. If so, novel therapies that treat high FGF23 levels, such as burosumab, may improve outcomes for patients with this treatable trait.

The direct effects of FGF23 on the innate immune system have yet to be fully investigated. There is a paucity of evidence on the effect of FGF23 on the effector function of the innate immune system in critically ill patients. Further investigation is needed to determine if FGF23 represents merely a marker of disease severity or an underlying mechanism of immune dysregulation. Likewise, more investigation is needed to establish whether the effects of changes in FGF23 could explain conflicting results in clinical trials of vitamin D replacement. Lastly, additional research may clarify whether antibody blockade of FGF23 augments the efficacy of vitamin D supplementation in patient response and whether FGF23 blockade decreases non-vitamin-D-dependent FGF23 effects on the innate immune response. 

Limitations: This study has a number of limitations: First, while both trials recruited patients with critical illness and had similar timepoints for sample analysis, the analysed samples were collected from different trials at different times, so inclusion and exclusion criteria were not uniform. Second, the dosing regimens for vitamin D supplementation were different, which may have affected patient responses. Third, different ELISA kits were used for sample measurement, which may have introduced error. Finally, samples were subjected to different lengths of storage time (+/− freeze–thaw cycles). Nevertheless, since the goal was to compare associations and trends, rather than absolute FGF23 levels, these limitations should not impact the data collected nor interfere with the results. To mitigate the differences between trials, we have analysed results separately and made conclusions about individual trials only. Our conclusions regarding the relationship between mortality and FGF23 are consistent across both studies despite different protocols and varying presentations of critical illness, suggesting further investigation is warranted. 

## 5. Conclusions

This study identifies FGF23 as a potential biomarker for patient outcomes in both the early and late stages of critical illness. The lack of relationship between FGF23 and vitamin D supplementation suggests that the normal regulatory mechanisms governing FGF23 production in response to vitamin D status are dysregulated in the development of critical illness. Further work is required to better understand the biology of FGF23 in critical illness and to inform the design of clinical trials on FGF23 pathway blockade. This translational work would pave the way for future trials investigating the possibility of FGF23 therapeutic manipulation through the use of inhibitory antibodies. 

## Figures and Tables

**Figure 1 biology-12-00309-f001:**
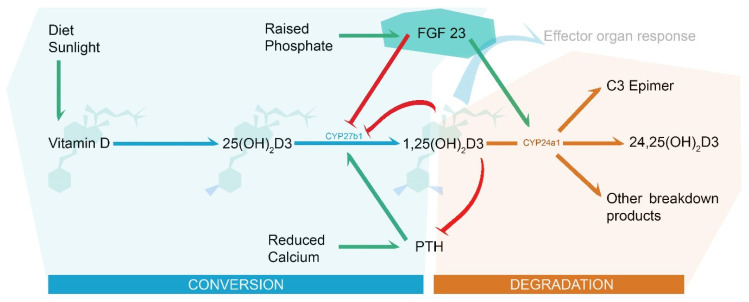
Vitamin D metabolism and the role of FGF23 (highlighted in turquoise). Green arrows indicate stimulus effect and may represent several enzymatic sub-steps, blue arrows indicate enzyme conversion to a functional 1,25(OH)2D3, orange arrows indicate degradation, and red arrows indicate downregulation/enzymatic blockade.

**Figure 2 biology-12-00309-f002:**
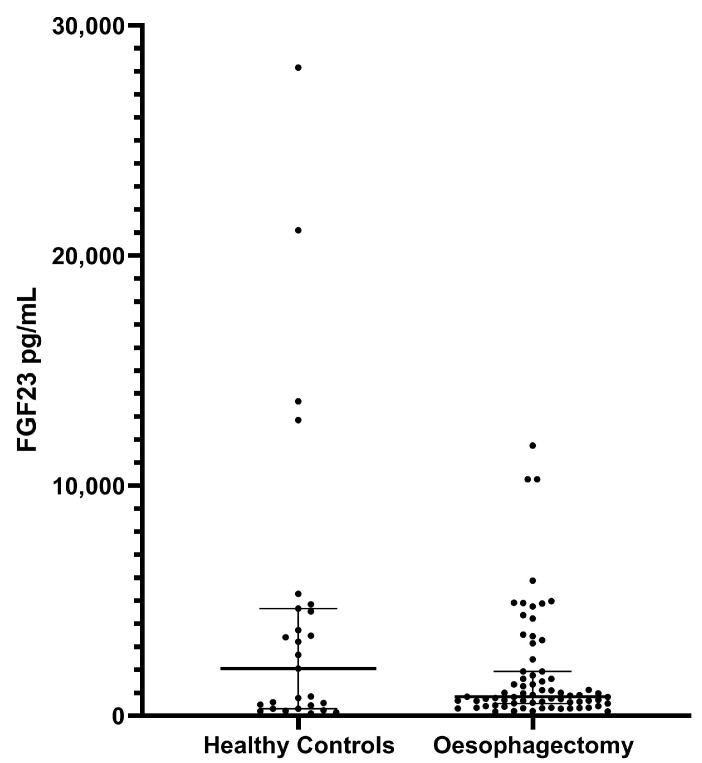
FGF23 levels in healthy controls vs. oesophagectomy patients. The median FGF23 (IQR) was 2057 pg/mL (307.1–4663.0) in healthy controls and 835.0 pg/mL (526.0–1929.0) in oesophagectomy patients (*p* = 0.4802). Critical illness patients were not included due to differences in FGF23 measurement protocols.

**Figure 3 biology-12-00309-f003:**
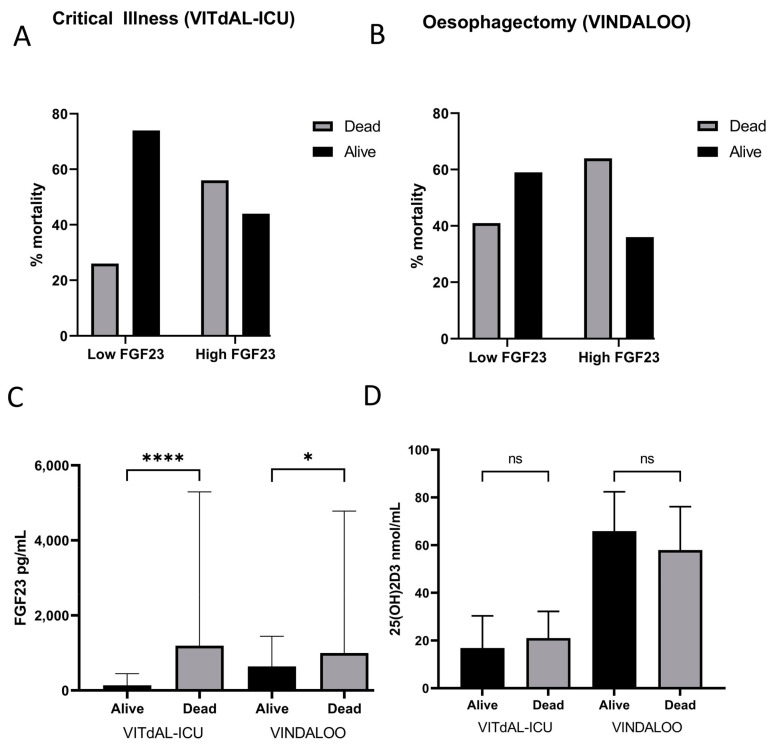
(**A**). Mortality of critical illness patients (VITdAL-ICU) at 6 months, based on their baseline (day 0) FGF23 levels. High baseline FGF23 levels resulted in 56% mortality, while low baseline FGF23 levels resulted in 26% mortality (chi-squared test with Yates correction, *p* < 0.0001). FGF23 cut-off was at 334 pg/mL (Q4). The grey bars represent the % of patients deceased after 6 months, while the black bars represent the % of patients still alive. (**B**). Mortality of oesophagectomy patients (VINDALOO) at two years, based on their pre-operative FGF23 levels. High FGF23 levels resulted in 64% mortality, while low FGF23 levels resulted in 41% mortality (chi-squared test with Yates correction, *p* = 0.0149). FGF23 cut-off was at 1104 pg/mL (Q4). The grey bars represent the % of patients deceased after two years, while the black bars represent the % of patients still alive. (**C**). baseline FGF23 levels of deceased vs. surviving patients in the critical illness (VITdAL-ICU) cohort (alive: median 136.2 pg/mL, IQR 61.3–450.3; deceased: median 1195 pg/mL, IQR 174.0–5295; *p* = 0.0462) and the oesophagectomy (VINDALOO) cohort (alive: median 641 pg/mL, IQR 332.0–1444; deceased: median 999 pg/mL, IQR 603.5–4784; *p* < 0.0001). Log-converted FGF23 levels were used to calculate significance. Data are represented as median and IQR. (**D**). Mean 25(OH)2D3 levels in alive vs. deceased groups at day 0 for both cohorts. No significant difference was observed between the alive and deceased groups for either cohort (critical illness *p* = 0.8274; oesophagectomy *p* = 0.6056). Data are represented as median and IQR, * *p* < 0.05, **** (*p* < 0.001), ns = not significant.

**Table 1 biology-12-00309-t001:** Analysis of the demographic results and mortality data of the cohorts. The mortality endpoint was 180 days for the critical illness cohort and 2 years for the oesophagectomy cohort. Mortality data were only available for 60 of the oesophagectomy patients. FGF23 levels were log converted for statistical data analysis. *p*-values were calculated using log-converted FGF23 levels. ^$^ Pre-critical-illness FGF23 levels (baseline FGF23) were not possible to collect as patients were not recruited before critical illness. CKD (chronic kidney disease).

	Critical Illness		Oesophagectomy		Healthy Controls
Total Patients	123		76		27
No. patients treated with vitamin D	62		40		NA
Male (%)	54.4%		84.2%		40.7%
Median age (IQR)	68.0 (56.0–77.0)		67.0 (58.0–71.5)		70 (61.0–78.0)
**Comorbidity at Recruitment**					
CKD 1–4 (%)	33 (27%)		2 (2.6%)		4 (15%)
Chronic liver disease	52 (42%)		1 (1.3%)		0 (0%)
Diabetes	7 (6%)		6 (7.9%)		6 (22%)
**Baseline FGF23**					
Median overall FGF23 (pg/mL)	NA ^$^		835.0		2057.0
IQR	NA ^$^		526.0–1929.0		307.1–4663.0
**Died (Baseline FGF23)**	**79/121**		**30/60**		**NA**
Median FGF23 (pg/mL)	1194.6		999		NA
IQR	176.9–5156.8		626.3–4659		NA
**Survived (Baseline FGF23)**	**42/121**		**30/60**		**NA**
Median FGF23 (pg/mL)	120.4		641		NA
IQR	55.9–392.1		345–1284		NA
	**Day 0**	**Day 7**	**Baseline**	**Day 3/4**	**NA**
FGF23 samples	123	122	75	59	NA
FGF23 (pg/mL)	202.3	171.9	879	1022	NA
FGF23 (lg10)	2.51	2.45	3.11	3.01	NA
Significance between baseline and timepoint 2	*p* = 0.6117		*p* = 0.7548		NA

**Table 2 biology-12-00309-t002:** FGF23 and interquartile range (IQR) for the critical illness cohort (VITdAL-ICU). D0 (day 0, baseline) and D7 (day 7, timepoint 2) show a significant difference between patients with CKD and those without (D0 *p* = 0.0001; D7 *p* = 0.0026). On D0: *n* = 33 patients with CKD and *n* = 90 without CKD. On D7: *n* = 31 patients with CKD and *n* = 91 without CKD. IQR (interquartile range).

Critical Illness				
	D0		D7	
	Median FGF23 (pg/mL)	IQR	Median FGF23 (pg/mL)	IQR
Overall	202.3	66.9–1217.0	171.9	62.4–610.4
No CKD	139.4	50.0–450.1	168.4	55.9–426.7
CKD	245.7	334.0–9334.8	182.3	157.4–6520.0

**Table 3 biology-12-00309-t003:** Timepoint 2 (day 3/4 for oesophagectomy patients; day 7 for critical illness patients). Comparison of FGF23 levels in treated vs. untreated patients with vitamin D replacement. No significant difference was observed between patients treated with vitamin D and those treated with placebo.

	Treated		Untreated		*p*
	Median FGF23 (pg/mL)	IQR	Median FGF23 (pg/mL)	IQR	
Oesophagectomy	1342	474.0–1754.0	819	573.4–2626.3	0.4053
Critical illness	211.8	56.7–458.0	120.4	66.1–1123.3	0.1424
No CKD	146.2	55.0–350.65	107.4	60.0–539.8	0.4483
CKD	296.2	102.5–3254.2	1406.7	478.1–8451.0	0.1345

## Data Availability

The datasets used and analysed during this study are available from the corresponding author on reasonable request.

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
