# Peer review of "Raised FGF23 Correlates to Increased Mortality in Critical Illness, Independent of Vitamin D"

_biology, 2023, doi:10.3390/biology12020309_

Round 1
Author Response
Thank you for your comments, we have made the necessary changes. Please see attached cover letter for detailed response.

Reviewer 2 Report
The main purpose of this manuscript is to draw attention to the clinical observation that high concentrations of FGF23 in blood are associated with an increased risk of dying in critically ill patients with different diseases, and this relationship is independent of vitamin D status. Because previous investigations of FGF23 have concentrated on its role in enhancing renal phosphate excretion and its inhibitory effect on renal 1-hydroxylation of 25-hydroxyvitamin D, the observations reported here add to the knowledge base of the complexities of the biology of FGF23. There are a number of points that need to be addressed by the authors:
1. Line 34 in the Abstract: “This was independent of vitamin D concentration,….” What is presumably meant here is “vitamin D status” as no measurements of the concentration of vitamin D in any material were reported.
2. Lines 110-111: “vitamin D level ≤20ng/ml.” There is no indication that vitamin D concentration was measured. Presumably this statement was meant to indicate that the concentration of 25-hydroxyvitamin D in blood plasma or serum was ≤20ng/ml. The substance being measured should be stated as 25-hydroxyvitamin D.
3. Figure 3 A and B: These histograms are confusing. For both low and high FGF23 the combined patients who died and patients who remained alive had 100% mortality. Furthermore, the grey portion of the histogram bars was stated to be the percentage of patients that were alive, yet the bar with high FGF23 indicates that a higher proportion of patients with high FGF23 remained alive than patients with low FGF23. This is in conflict with the main conclusion of this study. The meaning of Figure 3A and 3B needs to be clarified.
4. Line 333: “vitamin D level” This wording is incorrect as no concentration of vitamin D was measured. Presumably what is meant is “vitamin D status”.
Author Response
Thank you for your review, we have made the necessary changes. Please see the attached cover letter for a detailed response.

Round 2
